# Examining the Effect of Context, Beliefs, and Values on UK Farm Veterinarians’ Antimicrobial Prescribing: A Randomized Experimental Vignette and Cross-Sectional Survey

**DOI:** 10.3390/antibiotics10040445

**Published:** 2021-04-15

**Authors:** Sarah E. Golding, Jane Ogden, Helen M. Higgins

**Affiliations:** 1School of Psychology, Faculty of Health and Medical Sciences, Stag Hill Campus, University of Surrey, Guildford GU2 7XH, UK; j.ogden@surrey.ac.uk; 2Institute of Infection, Veterinary and Ecological Sciences, University of Liverpool, Neston, Cheshire CH64 7TE, UK; H.Higgins@liverpool.ac.uk

**Keywords:** antimicrobial prescribing, antimicrobial stewardship, farm animal medicine, beliefs, values, treatment decisions

## Abstract

Antimicrobial resistance (AMR) is a pressing threat to public and animal health. There is evidence that antimicrobial prescribing and stewardship behaviors by veterinarians (vets) are influenced by non-clinical factors, such as psychological, social, and environmental factors. This study explored the role of context, beliefs, and values on vets’ antimicrobial prescribing decisions. UK-based practicing farm vets (*n* = 97) were recruited to an online study. Using an experimental vignette methodology, vets were randomly assigned across four conditions, to examine the effects of different contexts (pressure on farm economics, the farmer, or the vet-farmer relationship, compared to a control condition) on vets’ likelihood of prescribing antibiotics. Vets’ beliefs about different groups’ responsibility for causing and preventing AMR and vets’ values were also measured. Key findings were that context alone, values, and beliefs about groups’ responsibilities for causing AMR were not predictive of vets’ likelihood of prescribing antibiotics. However, vets’ beliefs about groups’ responsibilities for preventing AMR were predictive of an increased likelihood of prescribing antibiotics, when vets were exposed to the experimental condition of the vignette in which the vet–farmer relationship was under pressure. Farm vets also believed that different groups have different levels of responsibility for causing and preventing AMR. Results should be interpreted cautiously, given the smaller than planned for sample size, and the possibility for both false negatives and false positives. Further research is needed to explore how these findings could inform antimicrobial stewardship interventions in veterinary medicine.

## 1. Introduction

Antimicrobial resistance (AMR) represents a profound threat to human and animal health [1]. Although AMR is a natural evolutionary process, increases in the rates of resistance amongst microorganisms are driven partly by inappropriate antimicrobial prescribing by doctors, veterinarians (vets), and other prescribers [2,3,4,5]. Tackling AMR will, therefore, involve developing antimicrobial stewardship interventions that enable prescribers across both human and veterinary medicine to reduce levels of inappropriate prescribing.

There is growing evidence that prescribers are influenced by a range of psychological, social, and environmental factors, although the volume of evidence varies, with a relative paucity of evidence about vets’ prescribing compared to doctors’ prescribing. Vets and doctors are influenced by psychological factors, such as emotions [6,7,8], habit [9,10], and risk perceptions [11,12], as well as by social factors such as interactions with their clients or patients [11,13,14]. Factors in the physical and social environment also influence prescribing, such as poor infection prevention and control (IPC) or biosecurity measures [15,16,17], and local and national policies [18,19,20,21].

Developing effective stewardship interventions, therefore, requires a good understanding of the various non-clinical factors that influence prescribing. In human medicine, a large number of studies using a range of methods have extensively researched and reviewed non-clinical influences on the prescribing behaviors of general practitioners [22,23], hospital doctors [11,24], and other prescribers [25,26]. In contrast to this substantial evidence base in human medicine, there have been fewer studies exploring the non-clinical aspects of prescribing in veterinary medicine. Until recently, much of the published research exploring prescribing in veterinary medicine has focused either on patterns of use [27,28,29,30] or on clinical factors influencing prescribing, such as underlying animal health or the use of antimicrobial susceptibility testing [31,32,33]. In those empirical studies that have explored non-clinical factors, much of the emphasis has been on areas of high antimicrobial usage in veterinary medicine, such as prescribing for pigs [17,34,35,36] or prescribing to treat or prevent udder diseases in dairy cows [37,38,39]. What evidence does exist about vets’ prescribing is, therefore, more tentative than the evidence for doctors’ prescribing, and it is important to continue building this evidence base to strengthen knowledge of non-clinical influences on vets’ prescribing. Furthermore, there are increasing calls for this work to be informed by the social sciences [14,40,41].

It is possible to experimentally explore the effects of different factors on prescribing using a vignette methodology [42,43]. Vignette studies are not without their limitations, but they do enable researchers to explore potentially influential factors in situations where a field-based experiment may be challenging to conduct [42,43]. In human medicine, vignettes have been widely used, and within the context of AMR have been specifically used to investigate antimicrobial prescribing decisions by GPs and hospital doctors across a range of conditions [44,45,46]. The use of vignettes within veterinary medicine is relatively limited however, although one recent vignette study identified a number of non-clinical factors that were associated with likelihood of prescribing antibiotics by farm vets, including habit, farmer relationship, and prescribing practices of other vets [10]. This current study, therefore, sought to contribute to the growing evidence base about non-clinical influences on veterinary prescribing decisions by applying a methodology (experimental vignettes) that is relatively under-utilized within veterinary medicine, to explore the influence of context, beliefs, and values on vets’ antimicrobial prescribing decisions. The rationale for selecting these factors will now be discussed.

### 1.1. Context

It is clear is that veterinary antimicrobial prescribing takes place in a complex and dynamic context, which can affect decision-making. For example, a vet’s prescribing can be influenced by economic factors [21,47,48], farm management practices [17,34,49], beliefs about clients [14,21,47,48], the nature of the vet–client relationship [13,37,47], and societal factors, such as policies [18,19,20,21]. With the exception of Doidge et al. (2019) [10], previous studies have only used vignettes to assess the appropriateness of vets’ antimicrobial prescribing in different clinical situations [50,51], rather than experimentally assess non-clinical or contextual aspects of the consultation scenario.

This study was, therefore, designed to further understand the relative importance of some contextual factors, informed specifically by previous qualitative research in which farm vets reported they feel prevented from always practicing ‘ideal’ antimicrobial prescribing due to three key contextual factors: situational factors on the farm (especially economics), farmer variability in personality and ability, and concerns for relationship management with their farmer clients [47]. These factors were operationalized in vignettes for this study to quantitatively explore which of these different elements of context (if any) had the greatest influence on vets’ likelihood of prescribing antimicrobials.

### 1.2. Beliefs

Previous research conducted with vets (as well as doctors, farmers, and other groups) has identified the phenomenon of other-blaming for the problem of AMR, with vets generally expressing beliefs that their own prescribing behaviors contribute less to the issue of AMR than prescribing by other vets and doctors or usage by farmers and the public [13,20,35,47,52]. This other-blaming may contribute to the sense of ambivalence towards stewardship expressed by vets and farmers [47]. In the qualitative study that directly informed this current study, vets and farmers attributed responsibility for both causing and preventing AMR across a number of different groups, including other vets and farmers, doctors, and the public [47].

This study, therefore, sought to measure farm vets’ beliefs about the level of responsibility that they, and other groups, have for both causing and preventing AMR. This was for two reasons; first, to assess whether such other-blaming would be evidenced again in a larger sample, using a different methodological approach, and second, as a way of attempting to quantify this other-blaming. Furthermore, given that those who engage in other-blaming may have reduced motivation to change their own behavior, as they do not perceive their own behavior to be a contributing factor to the problem of AMR, this study also sought to assess whether these beliefs about responsibility would be associated with vets’ likelihood of prescribing antimicrobials.

### 1.3. Values

The inappropriate use of antimicrobials can be considered a classic tragedy of the commons problem [53,54]; a commons problem arises in situations where commons resources, such as water or pasture, are exploited and misused by individuals, as such use is in their own best interests, even if there is a potential cost to others [53]. In this respect, AMR has been likened to another global challenge, that of climate change [40,54,55]. Successfully tackling climate change and AMR both require individuals to act in a way that will likely result in an immediate cost to them, with the benefits being realized by individuals and societies in different parts of the world, by the planet’s ecosystem, and by future generations [55]. Additionally, similar to climate change, the threat posed by AMR is considered by vets, doctors, and others to be a future threat, that is likely to affect others, rather than an immediate threat that could affect themselves [11,47,56]. Due to this similarity in these challenges, insights from environmental psychology related to environmental behaviors and values have been drawn upon in developing this study. Furthermore, since this study was conducted, others have also highlighted the potential role of values in influencing vets’ and doctors’ antimicrobial prescribing and stewardship behaviors [40,57].

Environmental behaviors have been defined as “all types of behavior that change the availability of materials or energy from the environment or alter the structure and dynamics of ecosystems or the biosphere” [58]. The use of antimicrobials can, therefore, be conceptualized as an environmental behavior, as the microbial environment has undoubtedly been influenced by human use of antimicrobials [59,60,61]. Pro-environmental behaviors are those that minimize harm to, or benefit, the environment [58]; antimicrobial stewardship can therefore be conceptualized as a pro-environmental behavior.

Research has demonstrated that four values (egoistic, hedonistic, altruistic, and biospheric) have been consistently associated with environmental beliefs, intentions, and behaviors [62,63,64,65,66,67]. Both hedonic and egoistic values reflect self-enhancement (or pro-self) values, and a concern with one’s own interests [63,68]. Hedonic values represent an individual’s concerns about reducing effort and increasing pleasure and comfort. Egoistic values represent an individual’s concerns about the costs and benefits of their actions in relation to safeguarding or increasing their own resources, including material goods and status. In contrast, altruistic and biospheric values reflect self-transcendence (or pro-social) values, and a concern with collective interests over self-interests [63,68]. Altruistic values represent concerns about improving or prioritizing the welfare of other human beings, while biospheric values represent concerns about preserving and protecting nature and the environment. Values are described as guiding principles for peoples’ lives [68] and while most people will endorse most values, they will generally differ in how they prioritize different values as being more or less important to them [62,69,70]. People who endorse pro-self values (egoistic and hedonistic) more than they endorse pro-social values (altruistic and biospheric) are less likely to engage in pro-environmental behaviors, such as recycling, energy reduction, or lower car use [63,66,67].

When people experience a situation in which their different values conflict, they are likely to act in accordance with the more salient value, which is usually the value that is more strongly endorsed [62]. It was, therefore, reasoned that differences in values between vets might partially explain some of the ambivalence towards stewardship that has been identified in previous research [47]. For example, if vets perceive AMR to be a threat to the collective health of others, then those vets who more strongly endorse altruistic values, and are more concerned with acting for the benefit of others than themselves, might exhibit more stewardship behaviors, especially in ambiguous clinical situations. Equally, if vets perceive AMR to be an environmental threat, then those vets who more strongly endorse biospheric values might exhibit more stewardship behaviors. In contrast, vets who more strongly endorse hedonic or egoistic values might choose to resolve challenging consultations by prescribing antimicrobials, and, therefore, exhibit fewer stewardship behaviors. To the best of the authors’ knowledge, the influence of values on vets’ antimicrobial prescribing and stewardship behaviors has yet to be explored empirically [40]. This study, therefore, sought to measure vets’ values and assess whether they were associated with their likelihood of prescribing antimicrobials.

### 1.4. Aims and Hypotheses

To summarize, three insights from previous research with vets have informed this current study. Firstly, vets report being influenced in their prescribing decisions by contextual factors. Secondly, vets hold differing beliefs about who is responsible for causing and preventing AMR and they engage in other-blaming for AMR. Thirdly, vets exhibit ambivalence towards tackling this issue, which may further influence their prescribing decisions and stewardship behaviors. Finally, it is also proposed that vets’ values may be associated with their prescribing.

The primary aim of the study was, therefore, to use an experimental vignette to explore whether farm animal vets’ antibiotic prescribing decisions were influenced by context. Secondary aims were to explore whether 1) vets held different beliefs about how responsible different groups were for causing and preventing AMR and 2) whether vets’ prescribing was also influenced by their beliefs about who has a responsibility for causing and preventing AMR, and by their values. The experimental hypotheses were:

#### 1.4.1. Vignette Condition (Context)

The primary hypotheses predicted that vets’ prescribing would be influenced by the context of the decision.

Hypothesis 1a. Vets provided with contextual information will be more likely to prescribe antibiotics, compared to the control condition (clinical information only provided);Hypothesis 1b. The likelihood of prescribing antibiotics will differ between the three different contextual (experimental) conditions. The four vignette conditions were control (clinical information only), economics (farm under financial pressure), farmer (farmer under work pressure), and relationship (vet-client relationship under pressure).

#### 1.4.2. Beliefs about Responsibility for Causing and Preventing AMR

Secondary hypotheses predicted that vets would believe different groups had different levels of responsibility for causing and preventing AMR, and that these beliefs would be related to their prescribing.

Hypotheses 2a, 2b. There will be a difference in the level of responsibility that vets believe different groups have for (a) causing AMR and (b) preventing AMR;Hypotheses 2c, 2d. Vets’ beliefs about which groups have responsibility for (c) causing AMR and (d) preventing AMR will be associated with vets’ antibiotic prescribing.

#### 1.4.3. Values

Additional secondary hypotheses predicted that vets’ prescribing would also be associated with the values they more strongly endorsed.

Hypotheses 3a, 3b. Vets who are (a) more hedonic and (b) more egoistic will be more likely to prescribe antibiotics;Hypotheses 3c, 3d. Vets who are (c) more altruistic and (d) more biospheric will be less likely to prescribe antibiotics.

## 2. Results

Participants were 97 farm animal vets from across the UK. Gender and ethnicity were disclosed by 91 participants (female = 46; male = 45; all described themselves as white). The age of 80 participants ranged from 23 to 69 years (median (*mdn*) = 33, interquartile range (*IQR*) = 12); in total, 17 participants declined to provide their age. The year in which they qualified was reported by 91 participants, who had been qualified for between 1 and 44 years (*mdn* = 10, *IQR* = 13). See Table 1 for full details of gender, ethnicity, and postgraduate qualifications. Participants were asked where in the UK they worked (Figure 1) and how often they worked with a range of animals; the majority reported that they often work with cattle and sheep, but less often with other species (Figure 2).

The survey link was accessed by 132 participants; in total, 17 did not progress beyond the information sheet and consent form, and 18 completed demographic information only, leaving 97 participants who completed at least the vignette (see Figure 3 for randomization to conditions and participant numbers completing subsequent measures). The values scales were completed by 92 participants, the responsibility for causing AMR scales were completed by 89 participants, and the responsibility for preventing AMR scales were completed by 88 participants. The target sample size was not reached within the planned six-month recruitment period, but recruitment ceased at the end of this period as all feasible avenues of recruitment had been exhausted during this time. Of the 88 participants who completed the study in one session, the mean time to complete the study was 15 min.

### 2.1. Distribution and Baseline Checks

Baseline differences were assessed using Fisher’s exact test or Pearson’s chi-square test for categorical variables and using one-way independent ANOVAs or the Kruskal–Wallis test for continuous variables. The associated *p*-values suggest there were no meaningful differences in participant characteristics between conditions at baseline on any variable, except for perhaps egoistic values (egoistic *p* = 0.062, *S* = 4.01; all other *p*-values ranged between 0.11 (*S* = 3.18) and 0.97 (*S* = 0.04); see Appendix A), suggesting the randomization to conditions was successful.

To confirm that demographic variables did not influence prescribing, a total of six bootstrapped hierarchical multiple regressions were conducted, with vignette condition entered in block one and the demographic variable of interest entered in block two (age, gender, years qualified, postgraduate qualifications, animals commonly worked with, or primary region of work). The associated *p*-values for these models ranged from 0.26 (*S* = 1.94) to 0.75 (*S* = 0.42), suggesting prescribing was not associated with these demographic variables. See Appendix A for summary model statistics.

### 2.2. Impact of Vignette Condition on Prescribing

The first set of hypotheses related to whether vets’ prescribing would be influenced by the provision of contextual, non-clinical information about either the on-farm economics, the farmer, or the vet-farmer relationship being under pressure. The first experimental hypothesis (1a) was that contextual information would influence vets’ prescribing, such that participants in the three experimental conditions would be more likely to prescribe antibiotics compared to those in the control condition (clinical information only). The second experimental hypothesis (1b) was that the likelihood of prescribing antibiotics would also differ between the three different experimental conditions. These hypotheses were tested using a one-way independent ANOVA. Mean likelihood of prescribing antibiotics was highest in the relationship condition, followed by the farmer condition, and lowest in the control condition, but the associated *p*-value, *S*-value, and small effect size suggest there was no meaningful difference overall between conditions of the vignette on the likelihood of prescribing antibiotics, *F* (3,93) = 1.64, *p* = 0.19, *S* = 2.40, partial *η*^2^ = 0.05 (see Figure 4 and Table 2). Both null hypotheses regarding the impact of contextual information on prescribing (1a and 1b) were therefore retained; providing vets with additional, contextual, non-clinical information did not affect their likelihood of prescribing antibiotics.

### 2.3. Vets’ Beliefs about Groups’ Responsibility for Causing and Preventing AMR

The third and fourth experimental hypotheses (2a, 2b) were that vets would believe that different groups had different levels of responsibility for causing and preventing AMR, with higher scores indicating greater responsibility (see Table 3 for descriptive statistics). To explore potential differences in beliefs, two Friedman’s ANOVAs were conducted). The associated *p*-values suggest there were meaningful differences in vets’ beliefs about the level of responsibility between groups for both causing AMR, *χ^2^*(5) = 130.06, *p* < 0.001, *S* = 9.97, and preventing AMR, *χ^2^*(5) = 100.51, *p* < 0.001, *S* = 9.97. In terms of both causing and preventing AMR, participants believed that human medics had the most responsibility and companion animal vets had the least responsibility. Differences between beliefs about all groups were explored using pairwise comparisons (see Appendix A for full details).

Regarding cause beliefs, farm vets believed themselves to be less responsible for causing AMR than farmers, human medics, and the public/patients (all *p* < 0.001, *S* = 9.97; see Table 3). Participants also believed farmers, medics, and the public/patients were more responsible for causing AMR compared to companion animal vets and pet owners (all *p* < 0.001, *S* = 9.97). Based on the remaining associated *p*-values (and the confidence intervals around these means), there were no meaningful differences in responsibility for causing AMR between farmers, medics, and the public/patients, or between farm vets, companion animal vets, and pet owners, suggesting that participants believed the former three groups are equally more responsible for causing AMR, and the latter three groups are equally less responsible for causing AMR.

Regarding prevent beliefs, farm vets believed themselves to have more responsibility for preventing AMR compared to companion animal vets (*p* < 0.001, *S* = 9.97), but less responsibility for preventing AMR than human medics (*p* = 0.03, *S* = 5.06; see Table 3). Based on the associated *p*-values (and the confidence intervals around these means) participants did not believe their own levels of responsibility differed from those of farmers, public/patients, and pet owners. Beliefs about companion animal vets differed from all other groups (all *p* < 0.001, *S* = 9.97, except versus pet owners *p* = 0.008, *S* = 6.97), suggesting participants saw this group as having the least responsibility for preventing AMR when compared to the responsibilities of the other five groups.

Regarding beliefs about causing and preventing AMR, the null hypotheses that there would be no difference in vets’ beliefs about the level of responsibility each group had for either causing AMR (hypothesis 2a) or preventing AMR (hypothesis 2b) were, therefore, rejected. Farm vets did differ in their beliefs about which groups had greater or lesser responsibility for both causing and preventing AMR, with human medics believed to have the most responsibility and companion animal vets the least responsibility.

### 2.4. Impact of Beliefs on Prescribing

The fifth and sixth experimental hypotheses (2c, 2d) were that participants’ beliefs about farm vets’ and other groups’ responsibility for causing or preventing AMR would be associated with the likelihood of prescribing antibiotics. These hypotheses were tested using two bootstrapped hierarchical multiple regressions.

In the first hierarchical regression (*n* = 89; see Table 4), the associated *p*-values for the effect of condition alone on prescribing (model one) was not meaningful overall, *F* (3,85) = 1.81, *p* = 0.15, *S* = 2.74, although it should be noted that the relationship condition had *p* = 0.059, *S* = 4.08. Adding beliefs about which groups have responsibility for causing AMR (model two) improved the overall model somewhat, *F* (9,79) = 1.74, *p* = 0.095, *S* = 3.40, (Δ*R^2^* = 0.11, *p* = 0.14, *S* = 2.84), with small associated *p*-values for the relationship condition (*p* = 0.048, *S* = 4.38) and beliefs about the public/patients causing AMR (*p* = 0.022, *S* = 5.51). This model explained 16.5% of variance in the sample (7.0% in the population).

In the second hierarchical regression (*n* = 88; see Table 5), the associated *p*-values for the effect of condition alone on prescribing (model one) suggested that any effect was not meaningful overall, *F* (3,84) = 1.53, *p* = 0.21, *S* = 2.25, although the relationship condition was potentially meaningful, *p* = 0.068, *S* = 3.88. Adding beliefs about which groups have responsibility for preventing AMR (model two) improved the model, *F* (9,78) = 2.21, *p* = 0.030, *S* = 5.06, (Δ*R^2^* = 0.15, *p* = 0.031, *S* = 5.01), with this model explaining 20.3% of variance in the sample (11.1% in the population), and the associated *p*-values suggest model two is meaningful overall. In this model, based on the *p*-values, most of the measures of beliefs about different groups having responsibility for preventing AMR were not meaningful individual predictors, although beliefs about the public/patients causing AMR could be potentially important, *t* (78) = 1.87, *p* = 0.065, *S* = 3.94. The associated *p*-value for the relationship condition in this model also suggests it is an important predictor of prescribing, *t* (78) = 2.24, *p* = 0.028, *S* = 5.16. Taken together, these results suggest that when beliefs about all groups being responsible (or not) for preventing AMR were included in the model, exposure to the relationship condition was meaningfully associated with an increased likelihood of participants’ prescribing antibiotics.

Regarding beliefs about causing and preventing AMR, the null hypothesis that cause beliefs would not be associated with prescribing (hypothesis 2c) was therefore retained, but the null hypothesis that prevent beliefs would not be associated with prescribing (hypothesis 2d) was rejected. Participants were more likely to prescribe antibiotics in the relationship condition, once their beliefs about groups’ responsibilities for preventing AMR were taken into account.

### 2.5. Vets’ Values

As a group, vets endorsed altruistic values most strongly, and egoistic values least strongly (see Table 6). In line with previous literature [63,64,65] vets’ hedonic and egoistic values were positively correlated with each other (*r* = 0.45, *p* < 0.001, *S* = 9.97) and vets’ altruistic and biospheric values were also positively correlated (*r* = 0.50, *p* < 0.001, *S* = 9.97). Altruistic values did not correlate with hedonic (*r* = −0.08, *p* = 0.45, *S* = 1.15) or egoistic values (*r* = −0.17, *p* = 0.11, *S* = 3.18) and biospheric values did not correlate with hedonic (*r* = −0.02, *p* = 0.87, *S* = 0.20) or egoistic values (*r* = −0.12, *p* = 0.25, *S* = 2.00).

### 2.6. Impact of Values on Prescribing

The seventh and eighth experimental hypotheses (3a, 3b) were that vets who were more hedonic or more egoistic would be more likely to prescribe antibiotics. The ninth and tenth experimental hypotheses (3c, 3d) were that vets who were more altruistic or more biospheric would be less likely to prescribe antibiotics. These hypotheses were tested using a bootstrapped hierarchical multiple regression. In this regression (*n* = 92; see Table 7), the associated *p*-values for the effect of condition on prescribing (model one) continued to suggest this effect was not meaningful overall, *F* (3,88) = 1.94, *p* = 0.13, *S* = 2.94; however, as with models presented above, the *p*-value associated with the relationship condition was low, *p* = 0.050, *S* = 4.32. Adding the values subscales (model two) did not meaningfully improve the model, *F* (7,84) = 0.87, *p* = 0.53, *S* = 0.92 (although again, relationship may still be important, *p* = 0.065, *S* = 3.94). Therefore, vets’ likelihood of prescribing antibiotics was likely not influenced by their values and the null hypotheses for 3a, 3b, 3c, and 3d were retained.

## 3. Discussion

This experimental vignette study examined the impact of context on farm vets’ antimicrobial prescribing decisions in a hypothetical clinical scenario commonly faced by farm vets. Participants were randomly allocated to a control condition, that presented only clinical information, or to one of three experimental conditions: economics (farm under financial pressure), farmer (farmer under work pressure), and relationship (vet-client relationship under pressure). The study also measured vets’ beliefs about which groups have more or less responsibility for both causing and preventing AMR and measured vets’ values.

The key findings were that context alone did not appear to meaningfully influence the likelihood that vets would prescribe an antibiotic, nor did vets’ values or their beliefs about which groups have responsibility for causing AMR, at least in this sample of farm vets. Adding vets’ beliefs about which groups have responsibility for preventing AMR did, however, appear to result in a more meaningful overall model; furthermore, in this model, exposure to the relationship condition (but not the economic or farmer conditions) appeared to be an important predictor for increased likelihood of prescribing, suggesting that the prescribing context may be more influential for some people, depending upon their beliefs. In other words, once vets’ beliefs about responsibility for preventing AMR were taken into account, those vets who were exposed to a hypothetical situation where their relationship with the farmer may be under some pressure were more likely to prescribe antibiotics than vets exposed to clinical information only, a situation where the farm is under financial pressure, or a situation where the farmer is under work pressure.

These findings, therefore, suggest the nature of the vet’s relationship with their farmer client may be a particularly important factor in terms of antimicrobial prescribing and stewardship behaviors. What is not clear from the present study is exactly what it is about the relationship being under pressure that resulted in vets being more likely to prescribe antibiotics. It may be that when vets feel they have a better quality relationship with a farmer, they feel more able to resist prescribing antimicrobials in clinically ambiguous situations. Alternatively, it may be that if vets feel less secure in their relationship with a farmer, they are more likely to prescribe antimicrobials as they wish to avoid either a potentially challenging conversation with the farmer or avoid the risk of displeasing the farmer by refusing antimicrobials. Nonetheless, the findings suggest that the more comfortable or confident a vet feels in their relationship with a farmer, the more likely it is that they will engage in responsible antimicrobial prescribing.

Relationships between prescribers and their patients or clients do seem to be important influences on prescribers’ treatment decisions [10,11,14]. There is evidence from doctors and other vets that a desire to avoid possible conflict or upset in the consultation is a potential driver of inappropriate antimicrobial prescribing [22,37,47,50]. Protecting relationships and maintaining patient or client satisfaction appears especially important to those prescribers who work in private healthcare systems or those who are concerned about losing clients or patients to other service providers [11,13,21,22]. Some vets and doctors report experiencing pressure to prescribe antimicrobials and it is likely that, at least sometimes, these prescriptions are written in order to maintain good will between the prescriber and the patient or client [11,17,34,47,71]. Indeed, companion animal vets report that establishing a trusting relationship and being able to communicate clearly and confidently with clients about responsible antimicrobial prescribing is key to being able to challenge this pressure and expectation from clients [13,50,72].

It, therefore, seems reasonable to propose that enabling vets and doctors to more quickly establish good communication and trusting relationships with clients and patients could be useful in driving antimicrobial stewardship behaviors. Indeed, junior dairy vets have recognized the importance of rapidly building trust with farmers, and have expressed a desire for more senior vets to facilitate this development of trust between junior vets and farmers [37]. One way to achieve greater communication and trust in livestock farming might be through increasing the opportunities for collaborative working. For example, senior pig vets in the UK feel that in most cases their relationships with farmers are collaborative and these senior vets do not report experiencing as much pressure to prescribe antimicrobials as their junior colleagues report experiencing [34]. Furthermore, there is emerging evidence that collaborative working between vets, farmers, and other livestock professionals can be beneficial in developing relevant, farm-level antimicrobial stewardship policies based around changes to treatment and biosecurity protocols, as well as other herd health interventions [73,74,75,76]. Importantly, there is evidence that these collaborative interventions can achieve reductions in antimicrobial usage without harming production parameters or animal welfare [73,74].

In addition to exploring the effects of context on farm vets’ prescribing, this study also set out to explore farm vets’ beliefs about which groups hold more or less responsibility for both causing and preventing AMR. The results identified that farm vets believe that farmers, human medics, and the public (as patients, not as pet owners) have greater responsibility than themselves for causing AMR. In terms of preventing AMR, farm vets believe they have more responsibility than companion animal vets, but less responsibility than human medics. These quantitative findings support the conclusions from previous qualitative research that farm vets do take some ownership and level of responsibility for antimicrobial stewardship, but that they also shift the responsibility for AMR and further stewardship onto other groups [47]. Indeed, there is a growing body of evidence that locating the problem of, and blame for, AMR with other groups is common amongst prescribers and users of antimicrobials; companion animal vets, equine vets, farm vets, doctors, dentists, farmers, the public, and others all express the view that practice by themselves or their own group is more responsible and less of a contributor to AMR than practice by other groups [13,17,20,35,52,77,78,79,80,81,82,83,84]. For example, in one survey of companion animal, bovine, and equine vets in Australia, although 50% thought veterinary antimicrobial usage contributed moderately to AMR, over 60% thought their own usage made only a minimal contribution to AMR [20]. Another survey of vets in Australia found that livestock vets and companion animal vets differed in the extent to which they believed antimicrobial usage in their own sector contributed to AMR, with each group rating their own sector as making less of a contribution than the other sector [79]. Surveys that have compared views across different professions have also found evidence of prescribers locating greater responsibility for AMR with other groups. For example, in a survey of doctors, dentists, and vets practicing in Australia, doctors believed that prescribing in their primary workplace made a ‘moderate’ contribution to AMR, but dentists and vets felt prescribing in their primary workplace made only a ‘minimal’ contribution [78]. Another large survey in Germany found different groups hold differing levels of acceptance that their own behavior influences AMR, with 70% of hospital doctors agreeing that their prescribing can drive AMR, compared to 53% of vets and 37% of the general public; furthermore, each group pointed to other groups as the key place to aim interventions [52].

Reducing this other-blaming between different groups of antimicrobial prescribers and users may be an important area of intervention, potentially positively impacting on on stewardship behaviours of individuals, or at least on their motivation to critique and consider changing their behavior. For example, there is evidence that vets and farmers feel stigmatized and blamed for AMR by others [85,86], which suggests that more inclusive campaigns such as ‘Antibiotic Guardian’ may be more acceptable to vets and farmers [87]. As discussed above, and by others elsewhere [57], collaborative efforts are needed to tackle the threats from AMR and there is emerging evidence that collaborations between different specialties across human and veterinary medicine can drive positive change in antimicrobial prescribing and usage [88,89]. Furthermore, One Health approaches to disease surveillance and management may help to overcome the ‘siloed’ approaches to education and funding, as well as improve communication between sectors and industries [41,86]. By emphasising the One Health nature of the threat from AMR, and increasing collaboration between groups, this may encourage individuals and groups to shift from blaming others for the problem to engaging in joint efforts to find the solution.

Finally, this study also measured farm vets’ values and assessed whether these were associated with vets’ likelihood of prescribing antimicrobials. Values were measured in this study because values have been associated with the likelihood of engaging in a range of pro-environmental behaviors [62,63,64,65,66,67], and antimicrobial stewardship has been conceptualized here as a pro-environmental behavior. Against expectations, no evidence was found in this study for any association between the four values that were measured (hedonic, egoistic, altruistic, and biospheric) and the likelihood of prescribing by farm vets. It is important to report these null findings regarding values, as others have also proposed that values may be of importance to vets’ prescribing decisions [40]. With hindsight, however, it is perhaps not surprising that there was no association between values and prescribing, as these values appear to act indirectly on behavior, through beliefs and norms [62,66]. Furthermore, other psychological factors, such as emotion or social identity, are also associated with pro-environmental behaviors [90,91]. Values mostly direct behavior when they are made salient by situational cues [70] and it would seem unlikely that altruistic and biospheric values would be readily or commonly activated during vets’ everyday prescribing encounters. For values to drive behavior, individuals have to perceive a behavior as being aligned to a given value. If, for example, vets do not think of inappropriate prescribing in terms of impacting upon the environmental health of the planet, then biospheric values would not be a predictor of inappropriate prescribing. Future research could explore whether the values measured in this study do influence vets’ prescribing when these values are cued to be more salient [70]. Alternatively, research could explore the potential role of other values not measured in this study [40,57]. Additionally, the use of values to inform and frame messaging within antimicrobial stewardship interventions may be an effective way of applying psychological and sociological theory in the context of vets’ and doctors’ prescribing behaviors [40], but further work is needed to explore this possibility.

### Limitations

This study aimed to recruit farm vets from across the UK and this was achieved, but no claims are made for the representativeness of this sample for farm vets currently practicing in the UK; indeed, a comparison with 2019 data from the Royal College of Veterinary Surgeons (RCVS) [92] indicates this is not a representative sample (although these RCVS demographic data are for all vets, with no breakdown for farm vets, who represented around 4% of RCVS-registered vets in 2019). According to these 2019 data, 59% of vets in the UK were female, 3.5% were from a Black, Asian, or other minority ethnic background, and the average age of registered vets was 40 for female vets and 51 for male vets. In this study, amongst those participants who provided their age, gender, and ethnicity, the average age was 33 years old, 51% were female, and all described themselves as White. The vets recruited for this study were therefore younger on average than vets registered with the RCVS, with minority ethnic groups and females under-represented, especially given that RCVS data show that the proportion of female vets is greater amongst younger age groups [92]. The British Veterinary Association and the RCVS were approached for assistance with recruitment, to both increase the sample size and broaden the recruitment channels; permission to recruit via the formal mailing lists of these organizations was not granted, although support was offered via their social media channels. The majority of participants were recruited via Twitter or through email requests direct to veterinary practices that provided a farm animal service, so there will inevitably be limitations in terms of the representativeness of those potential participants to whom the study was advertised. Recruitment of a sufficient number of participants was also challenging for this study; all feasible avenues were exhausted during the six-month recruitment period but the target sample size of a minimum of 116 participants was not achieved.

It is likely that this study was under-powered to detect small effects, and that the smaller than anticipated sample size may account for the lack of evidence for an overall effect of condition alone on farm vets’ prescribing, or of beliefs about responsibility for causing AMR influencing prescribing (based on the norm of accepting *p* < 0.05 as the threshold for statistical significance). Nonetheless, as with any results based on a single inferential test with what is arguably an arbitrary threshold [93,94], there is a risk that these results represent a false negative, especially as the observed effect size is smaller than was anticipated and planned for following the pilot study, and given that the associated *p*-values for the relationship condition were consistently low across all models. Observed (*post hoc*) power calculations have not been performed, however, as it has been demonstrated that observed power is directly related to the observed *p*-value, and results with a *p*-value > 0.05 will necessarily have low observed power [95]. Calculating observed power does not provide any additional analytic benefit over and above the observed *p*-value and discussions of observed power in relation to non-significant results are not recommended [95,96].

Indeed, researchers are encouraged to avoid judgements about results based on statistical inference, and to not describe results as significant or otherwise based solely on *p*-values, but to consider the results in the context of what is already known about a topic [93,94]. In this light, and further to the previous discussion about the possibility of false negatives, it should be acknowledged that although multiple analyses were performed on this dataset, a Bonferroni adjustment was not applied; this means the apparently meaningful regression model that included beliefs about responsibility for preventing AMR could be a false positive. In assessing whether or not a Bonferroni adjustment should be applied, researchers are encouraged to consider the potential risks associated with accepting a false positive versus accepting a false negative [97,98]. The Bonferroni adjustment can be considered too conservative for much research, as the risk of accepting a false negative can include ignoring a potentially interesting or meaningful effect [97,98]. Given the findings about the potential importance of relationships and other-blaming that have been identified in the wider literature [10,13,20,35,37,47,52,56], it was deemed appropriate in this instance not to apply the Bonferroni adjustment and instead accept the model, but acknowledge the risk that the finding may be a false positive. Overall, these findings about the potential influence of relationships and beliefs on prescribing should be considered tentatively, and further studies should be conducted to explore whether these findings can be replicated in future vignette studies, ideally in larger samples and across different groups of prescribers.

Finally, it should be acknowledged that there may be differences between vets’ self-reported likelihood of prescribing antibiotics in response to the scenarios presented in the vignettes in this study and how they would actually behave if they faced these situations in practice. For example, it may be that social desirability bias influenced participants’ responses, or that participants interpreted the specific content of the vignette in a way that might differ during their everyday clinical practice. It is a recognized limitation of vignette methodology that participants’ responses in vignette studies may not fully reflect their real-world behavior [43], but there is nonetheless some evidence that supports a correlation between participants’ actual behavior in clinical practice and their responses in vignette studies [43,99,100]. Findings should therefore be interpreted in this light; responses to vignettes are only proxies for actual behavior, and these findings should be used to help guide future studies that will explore the influence of non-clinical factors on antibiotic prescribing decisions in everyday clinical practice.

## 4. Materials and Methods

### 4.1. Design

This between-participant online experimental study used vignettes to investigate the influence of different contextual information on farm vets’ likelihood of prescribing antimicrobials. Participants were randomly presented with one of four vignettes: the control condition provided only clinical information about a dairy cow’s presentation, the economics condition suggested the farm was under financial pressure, the farmer condition suggested the farmer was experiencing pressure at work, and the relationship condition suggested the vet–client relationship was under pressure from a previous negative interaction. Additionally, this study also collected cross-sectional data about farm vets’ values, and about their beliefs about which groups have responsibility for causing AMR and which groups have responsibility for preventing AMR; these measures were taken as additional potential predictors of prescribing. There was no follow-up element to the study.

### 4.2. Participants

Recruitment took place from January to June 2018. Participants were farm animal vets from across the UK who were recruited opportunistically by advertising the online study on Twitter, in the British Cattle Veterinary Association’s member e-newsletter, via email to members of the Sheep Veterinary Society, via email to veterinary practices with farm vets, through snowball sampling, and through the researchers’ professional veterinary and farming networks. Inclusion criteria were that participants spent at least some of their working week in clinical veterinary practice with livestock animals of any species. There was no upper age limit and no restrictions to participation on the basis of gender, ethnicity, or other demographic variable. Participants who completed the study were given the opportunity to enter into a draw to win one of two £25 shopping vouchers.

#### Power Calculation

A pilot study [101] explored whether vets’ self-reported prescribing behavior differed between two conditions: (1) when they were asked to think about clinical information only in ‘ideal, text-book’ situations and (2) when they were asked to consider their own experience of prescribing in ‘real-world’ situations. This study found that across the 7 clinical scenarios presented, vets were more likely to prescribe antibiotics in ‘real-world’ situations compared to ‘ideal’ situations. Based on this pilot study (in which effect sizes ranged from *r* = 0.36 to *r* = 0.52), it was estimated that a medium effect size (*r* = 0.3, partial *η*^2^ = 0.09) could be reasonably expected in this current vignette study. A power calculation conducted using G*Power [102] indicated a minimum sample size of 116 participants would be needed to detect an effect size of *r* = 0.3 using an ANOVA to test the effect of vignette condition. To test the effects of additional variables using multiple regression models, 192 participants would be sufficient to detect an effect of this size using a regression model with 13 predictors. Other rules of thumb used to guide sample size for regression models suggest a minimum of 10 participants per predictor is considered an absolute minimum, but 30 participants per predictor will offer better power [103]. The study, therefore, aimed to recruit between 116 and 192 participants.

### 4.3. Experimental Materials and Measures

#### 4.3.1. The Vignette

Participants were randomly assigned to read one of four versions of a hypothetical clinical scenario about a dairy cow who was displaying ambiguous clinical signs, but where the likely ‘correct’ diagnosis was infectious bovine rhinotracheitis (IBR). IBR is a viral infection for which prescribing antibiotics would usually be considered inappropriate. This scenario was developed and tested in a pilot study [101] and was selected for use in this current study as it resulted in the greatest mean difference in prescribing between ‘ideal’ and ‘real-world’ conditions, compared to other scenarios tested in the pilot study (mean difference represented by a large effect size, *r* = 0.52). Furthermore, this scenario represents genuine uncertainty for vets; clinically, the cow is likely suffering from a viral infection (IBR) that she has contracted from another animal in the herd, but it is possible the primary causal agent is bacterial or that she might develop a secondary bacterial infection. Finally, no participant in the pilot study responded that they would always prescribe antibiotics in this IBR scenario in the real-world. In contrast, for the only other vignette in the pilot study that demonstrated the same large effect size, (which related to the use of antimicrobial therapy at the point of drying off a dairy cow), several participants responded they would often or always prescribe antimicrobials, suggesting the potential for a ceiling effect had this scenario been used in this current experimental study, where subtle differences in the impact of context might not be detected.

The scenario was amended slightly from the pilot study to remove potential ambiguity in the vignette that could have affected vets’ decision-making; in the pilot study, the cow was described as a ‘dairy cow’, which was changed to ‘dry cow’ for this study. It was made clear that the cow was not currently giving milk (she was “dry”) for two reasons. First, when a cow is giving milk there is an economic argument not to prescribe antibiotics to avoid a milk withdrawal period (during which an antibiotic-treated cow’s milk cannot be sold), or to prescribe certain critically important antibiotics, such as third and fourth generation cephalosporins, as these products do not always carry a milk withdrawal period; this potential economic influence on prescribing decisions for lactating cows could have confounded the results by ‘contaminating’ all versions of the vignette. Second, it was important to make it clear to participants that the cow was not lactating, so as to avoid the possibility that some participants might imagine she was lactating, while others might imagine she was not lactating. The final version of the vignette was piloted again, with feedback sought from 4 farm vets and 7 psychologists.

The control version of the vignette contained only clinical information about the cow; this same clinical information was also presented in the other three conditions. The 3 experimental vignettes contained additional pieces of text that provided contextual information; this information was designed to describe a challenging context that vets have previously reported they can experience in everyday clinical practice [47]. The experimental conditions were: a challenging economic context (farm business under financial pressure; referred to above as the ‘economics’ condition), a challenging farmer context (farmer themselves under pressure at work; the ‘farmer’ condition), and a challenging relationship context (vet-farmer relationship under pressure; the ‘relationship’ condition).

All participants were instructed to “Please read the following paragraph and think about the possible treatment options you might recommend in this scenario”. The vignette read as follows:


*Imagine you have been called to a dairy farm to examine a sick dry cow. She is bright and alert, but has nasal discharge and a rectal temperature of 39.2 °C. You find no other clinical abnormalities. You are aware that there has been a recent laboratory confirmation of IBR (infectious bovine rhinotracheitis) in the herd, which was previously naive to IBR.*
[Control].

The experimental vignettes contained the following additional text, presented at the end of the clinical information:


*You are familiar with this farm business and know that it has been having real financial difficulties for quite some time now.*
[Economics].


*You know this farmer is well trained and usually diligent, but he says his herdsman who usually manages the cows day-to-day has recently resigned and has yet to be replaced. It is obvious that the farmer is run off his feet and keeping an eye on this cow is likely to be a low priority for him right now.*
[Farmer].


*You have known this farmer for a while and the last time you visited him it was quite tense, and he wasn’t happy with some of your decisions. As you are examining the cow, you hear the farmer grumbling.*
[Relationship].

#### 4.3.2. Outcome Measure: Antibiotic Prescribing Behavior

The outcome measure was how likely participants would be to prescribe systemic antibiotics in each condition. After reading the vignette, participants were presented with the following instructions:


*Below are some common treatment options that vets might recommend in this scenario. Thinking about your own day-to-day practice, please indicate to what extent you would be likely to recommend each of the following options in the above scenario.*


Participants then indicated for 7 treatment options how likely they would be to recommend each option, using a Likert scale from 1 (*very unlikely*) to 5 (*very likely*). The treatment options were (i) systemic antibiotics, (ii) cortico-steroids, (iii) isolation from other animals, (iv) provide highly palatable diet, (v) no treatment, (vi) non-steroidal anti-inflammatories, and (vii) other. The order in which the treatment options were presented in the list was randomized to control for any order effects.

Participants were then presented with two free text boxes and asked to respond to two statements: “Please briefly provide further details about any specific drug treatments you have recommended (e.g., dose, type, duration), or if you chose ‘other’ please elaborate” and “Please briefly describe your reasons for this course of action”. 

### 4.4. Cross-Sectional Measures

#### 4.4.1. Demographics

Participants were asked to provide their age, gender, and ethnicity; they were also asked about their veterinary experience (the year they qualified, whether they had postgraduate qualifications, how regularly they worked with certain animals, their job title, and their primary region of work).

#### 4.4.2. Values

Values were measured using the value orientation scale, which has been extensively used in environmental psychology and has been validated across different countries [63,64,65]. There are 16 items, comprising four sub-scales (hedonic, egoistic, altruistic, and biospheric). Participants are asked to “Please rate how important each value is for you as a guiding principle in your life” and to respond on a scale of -1 (*opposed to my values*) to 7 (*of supreme importance*), to indicate to what extent they endorse certain values (e.g., *equality*, *wealth*, *preventing pollution*, *pleasure*). For the data analysis, responses to these subscales were recoded as ranging from 1 to 9.

#### 4.4.3. Beliefs: Responsibility for Causing AMR and Responsibility for Preventing AMR

Beliefs about which groups of stakeholders have responsibility for causing AMR and responsibility for preventing AMR were measured using two 24-item scales that were developed for this study (see Appendix B). These scales were developed based on previous interviews with farm vets [47], which identified 6 key stakeholder groups that vets felt had responsibility for both causing and preventing AMR. Those six groups were farm animal vets, farmers, companion animal vets, pet owners, human healthcare professionals, and the public/patients. Each of the 2 scales comprised 6 subscales, with 4 items about each of the 6 groups. For both scales, participants were asked “To what extent do you think each of the following contributes to causing/preventing antimicrobial resistance?”. Participants responded on a scale of 1 (*contributes not at all*) to 5 (*contributes very much*). Both scales were piloted to check for clarity and meanings of items with the same vets and psychologists who piloted the vignette.

The responsibility for causing AMR scale measured the extent to which farm vet participants think the behavior of different groups (including themselves) contributes to increasing rates of AMR. Statements presented for each group included a mix of statements about antibiotic use, biosecurity, and infection prevention and control (IPC) practices. Example items that participants were asked to rate are: “The number of antibiotic prescriptions that GPs write”, “Levels of compliance with infection control protocols in companion animal veterinary practices”, and “Farmers using antibiotics to compensate for issues with husbandry”.

The responsibility for preventing AMR scale measured the extent to which participants think all groups are equally responsible for preventing AMR. Statements are focused on what members of each group could do differently to help reduce selective pressure for AMR. Example items that participants were asked to rate are: “Members of the public taking antibiotics as instructed by their doctors”, “Pet owners accepting that their animals don’t always need antibiotics”, and “Farm animal vets adopting antibiotic stewardship policies”.

#### 4.4.4. Reliability

Cronbach’s alphas (*α*) were calculated for all measures and are reported in Table 8. For the values measures, internal consistency was acceptable for all subscales. For the responsibility for causing AMR and responsibility for preventing AMR measures, 5 subscales did not have acceptable internal consistency; from each of these subscales, one item was removed to improve internal consistency. Following this, 2 subscales (the public and patient subscales in both the causing and preventing AMR scales) still did not quite meet the threshold for acceptable internal consistency of 0.6 for the early stages of research [104] but were nonetheless retained in the final analysis. New variables were created for each values and beliefs subscale, whereby the total score was divided by the number of items, so that mean scores could be comparable across the different values or beliefs subscales.

### 4.5. Procedure

All study materials were presented online using Qualtrics. After viewing the participant information sheet and completing the consent form, participants were asked to indicate how often they worked with different animals. Next, participants saw 1 of 4 versions of the experimental vignette, and were then asked to indicate which treatment recommendations they would be likely to make if they faced this scenario in real life. Participants then completed 3 scales to measure their values, their beliefs about which groups are responsible for causing AMR, and their beliefs about which groups are responsible for preventing AMR. Following these scales, participants were asked to provide brief demographic details. Finally, participants saw a debrief screen and were asked to provide an email address if they wished to be entered into the shopping voucher prize draw.

For transparency, it should be noted that minor changes were made to the information sheet, consent form, and demographics questionnaire (which was initially presented immediately after the consent form and before the vignette) around 2 weeks after the study was launched. The initial drop-out rate was 40%; participants were either not progressing beyond the consent form or were only completing some of the demographics information. In total, 3 key changes were made: the information sheet was edited to be more succinct, the consent form was altered so participants only had to agree to one global statement at the end of the form (instead of agree to a list of statements, plus the global statement), and the demographic questionnaire was moved to the end of the study. After these changes were implemented the drop-out rate reduced to 24%. 

#### Allocation and Randomization

All randomizations were performed using the Qualtrics software. Participants were randomly allocated to the control and experimental vignette conditions; randomization was not stratified or performed in blocks. As the design was between-subjects, allocation to condition was not revealed to participants; however, allocation to condition was not blinded to the experimenter during data analysis. To control for order effects, the order of treatment options presented after the vignette was randomized, as was the order of presentation for all scale items within each scale. Additionally, the order of presentation of the responsibility for causing AMR and responsibility for preventing AMR scales was also randomized, to control for any potential contamination between the 2 scales.

### 4.6. Data Analysis

Missing data were not imputed; cases with missing data were excluded pairwise in all analyses (except the hierarchical multiple regressions, where they were excluded listwise). The ‘forced entry’ option was used in Qualtrics for all items in the main measures, meaning that missing data only occurred when participants declined to continue the survey; there were no missing data for individual items on completed measures. Data analysis was conducted using IBM SPSS Statistics version 24.

Baseline checks of group differences (to check randomization to conditions) were performed using Pearson’s chi-square test and Fisher’s exact test for categorical variables and one-way independent ANOVAs and Kruskal–Wallis tests for continuous variables. The impact of context (vignette condition) on the likelihood of antibiotic prescribing was tested using a one-way independent ANOVA. Differences in beliefs about different groups’ responsibility for causing or preventing AMR were tested using Friedman’s ANOVAs, followed by pairwise comparisons. The impact of demographics, beliefs, and values on prescribing were tested using bootstrapped hierarchical multiple regressions. All regressions were conducted using the ‘Enter’ method and dummy variables were created for categorical variables.

To aid interpretation of results, in addition to reporting *p*-values, 95% confidence intervals, and effect sizes, the *S*-value (Shannon Information or binary surprisal) is also reported. The *S*-value is the base-2 logarithmic transformation of the *p*-value, which transforms the *p*-value into a measure of information (in bits; binary digits) against a test hypothesis; the greater the *S*-value, the less compatible the observed data are with the test hypothesis [105,106,107]. It is considered a more intuitive metric for interpreting evidence against the test hypothesis (in this study, the null hypotheses), as it transforms probabilities associated with test statistics into a measure of information that can be understood in terms of the amount of information that would be gained from the same number of coin tosses (by rounding the *S*-value to nearest integer) [105,106,107]. For example, an *S*-value of 2 (meaning there are 2 bits of information against the test hypothesis) indicates that the observed test statistic is about as surprising as tossing two heads in two fair coin tosses, whereas an *S*-value of 10 would indicate the test statistic is about as surprising as tossing 10 heads in 10 fair coin tosses.

### 4.7. Ethics

A favorable ethical opinion for the study was granted by the University of Surrey’s Ethics Committee (Reference: UEC/2017/105/FHMS). Participants were provided with an information sheet and written informed consent was provided by participants before commencing the study. Participation was voluntary and participants could withdraw at any point until they completed the survey; as data collected were anonymous, there was no mechanism for identifying and removing responses from individual participants. To enter the optional prize draw, participants provided an email address after they completed the survey; email data were stored separately from the study data. The study was not pre-registered but has been reported in line with guidelines for experimental and cross-sectional studies: the Consolidated Standards of Reporting Trials (CONSORT) checklist [108] and the Strengthening the Reporting of Observational Studies in Epidemiology Statement (STROBE) checklist [109].

## 5. Conclusions

The key findings from this online experimental vignette and cross-sectional survey were that, once their beliefs about various groups’ responsibility for preventing AMR were taken into account, farm vets exposed to a hypothetical scenario in which their relationship with a farmer was under pressure were more likely to prescribe antimicrobials than farm vets in other contextual conditions. Farm vets’ values and their beliefs about various groups’ responsibility for causing AMR were not associated with their likelihood of prescribing antimicrobials. This study also found that there were differences in farm vets’ beliefs about how responsible different groups are for causing and preventing AMR, adding to previous evidence that farm vets at least partially locate the problem of AMR and the responsibility for antimicrobial stewardship with other groups. Results should be interpreted cautiously given the smaller than planned for sample size and given the risk for both false negatives and false positives as discussed in the limitations. Further research is needed to explore whether these findings can be replicated in different samples, and to examine how these findings could inform antimicrobial stewardship interventions in veterinary medicine.

## Figures and Tables

**Figure 1 antibiotics-10-00445-f001:**
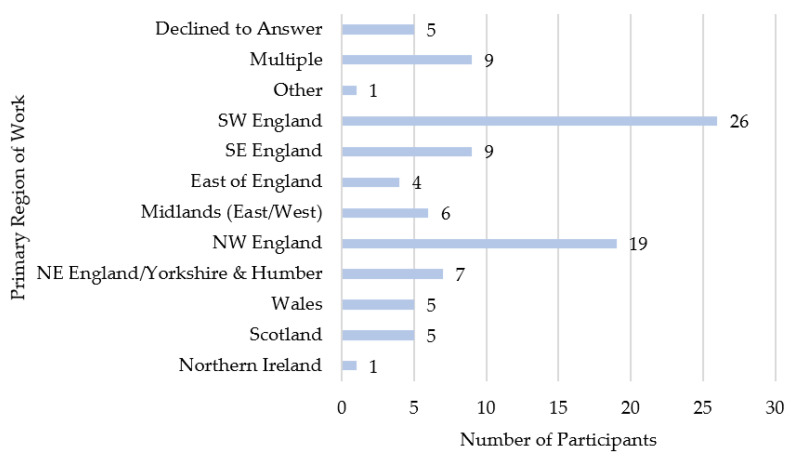
Participants’ primary region of work. NE = North East. NW = North West. SE = South East. SW = South West. For a map of UK regions, please refer to the Office for National Statistics website: https://geoportal.statistics.gov.uk/search?collection=Document&sort=name&tags=all(MAP_RGN) (accessed on 13 April 2021).

**Figure 2 antibiotics-10-00445-f002:**
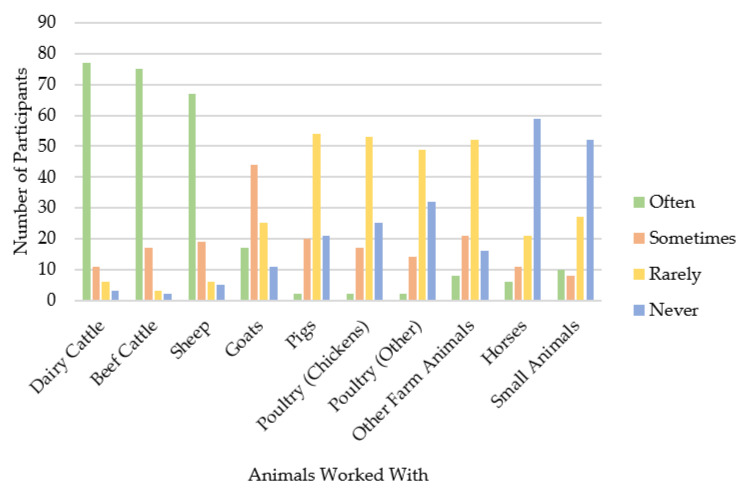
Participants’ responses to “How often do you look after the following animals?”.

**Figure 3 antibiotics-10-00445-f003:**
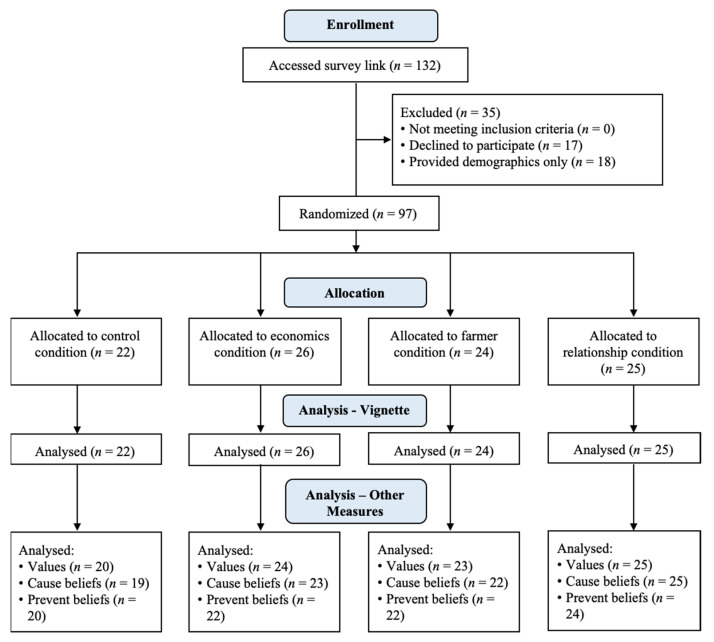
CONSORT (Consolidated Standards of Reporting Trials) diagram showing flow of participants through study.

**Figure 4 antibiotics-10-00445-f004:**
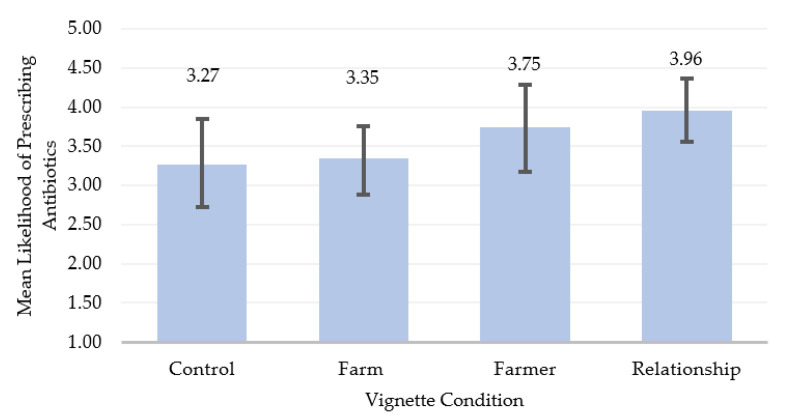
Mean likelihood of prescribing antibiotics by vignette condition (*n* = 97); response units represent 5-point Likert scale from 1 (*very unlikely*) to 5 (*very likely*). Error bars represent 95% bias-corrected and accelerated bootstrap confidence intervals.

**Table 1 antibiotics-10-00445-t001:** Demographic Details for All Vignette Participants.

Characteristic	Response Option	Individuals in Sample (*n* = 97)	Percentage of Sample
Gender **^1^**	Female	46	46.39
	Male	45	47.42
	Other	0	0.00
	Prefer Not to Say	1	1.03
	Declined to Answer	5	5.15
Ethnicity **^1^**	White	91	93.81
	Black	0	0.00
	Asian	0	0.00
	Mixed	0	0.00
	Other	0	0.00
	Prefer Not to Say	1	1.03
	Declined to Answer	5	5.15
Holds Postgraduate Veterinary Qualifications	Yes	26	26.80
	No	65	67.01
	Declined to Answer	6	6.19

^1^ Percentages do not exactly total 100% due to rounding.

**Table 2 antibiotics-10-00445-t002:** Mean Likelihood of Prescribing Antibiotics by Vignette Condition.

Condition	Mean	Lower BCa 95% CI ^1^	Upper BCa 95% CI ^1^	Standard Deviation
Control	3.27	2.73	3.85	1.39
Economics	3.35	2.88	3.76	1.23
Farmer	3.75	3.18	4.29	1.36
Relationship	3.96	3.56	4.36	1.06

^1^ BCa 95% CI = 95% bias-corrected and accelerated bootstrap confidence intervals. Bootstrap results are based on 1000 bootstrap samples.

**Table 3 antibiotics-10-00445-t003:** Beliefs about Groups’ Responsibility for Causing or Preventing AMR.

Beliefs	Subscale	Mean ^1^	Lower BCa 95% CI ^2^	Upper BCa 95% CI ^2^	Standard Deviation	*p*	*S*
Responsibility for Causing AMR (*n* = 89)	Human Medics	3.77	3.63	3.92	0.68	<0.001	9.97
	Public/Patients	3.72	3.56	3.88	0.75	<0.001	9.97
	Farmers	3.62	3.46	3.77	0.72	<0.001	9.97
	Farm Animal Vets ^3^	3.19	3.05	3.34	0.70	-	-
	Pet Owners	3.18	3.01	3.35	0.82	0.95	0.07
	Companion Animal Vets	3.05	2.90	3.21	0.73	1.00	0.00
Responsibility for Preventing AMR (*n* = 88)	Human Medics	4.28	4.14	4.41	0.62	0.031	1.69
	Farmers	4.21	4.07	4.35	0.65	0.072	3.80
	Public/Patients	4.08	3.94	4.22	0.64	1.00	0.00
	Farm Animal Vets ^3^	4.06	3.94	4.19	0.60	-	-
	Pet Owners	3.93	3.78	4.09	0.72	1.00	0.00
	Companion Animal Vets	3.63	3.48	3.79	0.74	<0.001	9.97

^1^ Response units for mean represent 5-point Likert scale assessing extent to which each group’s actions are believed to contribute to causing or preventing AMR from 1 (*contributes not at all*) to 5 (*contributes very much*); results presented in descending order of believed contribution.^2^ BCa 95% CI = 95% bias-corrected and accelerated bootstrap confidence intervals. Bootstrap results are based on 1000 bootstrap samples. ^3^ Farm vets designated as reference group.

**Table 4 antibiotics-10-00445-t004:** Hierarchical Regression Model for Beliefs about Responsibility for Causing AMR as Predictors of Prescribing.

	Variable	*B* [BCa 95% CI] ^2^	*SE B*	*Β*	*t*	*p*
Block 1	Constant	3.21 [2.51, 3.83]	0.30		10.87	<0.001
	Control v. Economics	0.05 [−0.75, 0.86]	0.40	0.02	0.13	0.90
	Control v. Farmer	0.52 [−0.45, 1.53]	0.40	0.17	1.28	0.20
	Control v. Relationship	0.75 [−0.01, 1.57]	0.39	0.26	1.91	0.059
Block 2 ^1^	Constant	5.11 [3.03, 7.15]	0.96		5.34	<0.001
	Control v. Economics	−0.006 [−0.77, 0.80]	0.40	−0.002	0.02	0.99
	Control v. Farmer	0.32 [−0.64, 1.27]	0.40	0.11	0.79	0.43
	Control v. Relationship	0.79 [−0.24, 1.55]	0.39	0.27	2.01	0.048
	Medics Causing AMR	−0.06 [−0.50, 0.44]	0.23	−0.0	0.24	0.81
	Public/Patients Causing AMR	−0.56 [−0.95, −0.13]	0.24	−0.32	2.33	0.022
	Comp. Animal Vets Causing AMR	−0.32 [−0.87, 0.37]	0.27	−0.18	1.18	0.24
	Pet Owners Causing AMR	0.22 [−0.23, 0.70]	0.24	0.14	0.92	0.36
	Farm Vets Causing AMR	0.43 [−0.27, 0.94]	0.35	0.23	1.24	0.22
	Farmers Causing AMR	−0.18 [−0.81, 0.60]	0.33	−0.10	0.53	0.60

^1^*R^2^* = 0.060 for Block 1; Δ*R^2^* = 0.105 for Block 2 (*p* = 0.14). ^2^ BCa 95% CI = 95% bias-corrected and accelerated bootstrap confidence intervals. Bootstrap results are based on 1000 bootstrap samples. Comp. = Companion. V. = Versus.

**Table 5 antibiotics-10-00445-t005:** Hierarchical Regression Model for Beliefs about Responsibility for Preventing AMR as Predictors of Prescribing.

	Variable	*B* [BCa 95% CI] ^2^	*SE B*	*Β*	*t*	*p*
Block 1	Constant	3.20 [2.54, 3.81]	0.29		11.16	<0.001
	Control v. Economics	0.12 [−0.71, 1.01]	0.40	0.04	0.30	0.77
	Control v. Farmer	0.53 [−0.28, 1.37]	0.40	0.18	1.33	0.19
	Control v. Relationship	0.71 [−0.006, 1.53]	0.39	0.25	1.85	0.068
Block 2 ^1^	Constant	6.15 [4.35, 7.90]	1.08		5.70	<0.001
	Control v. Economics	0.34 [−0.55, 1.33]	0.41	0.12	0.84	0.41
	Control v. Farmer	0.53 [−0.26, 1.33]	0.39	0.18	1.36	0.18
	Control v. Relationship	0.86 [0.05, 1.78]	0.38	0.30	2.24	0.028
	Medics Preventing AMR	0.14 [−0.52, 0.86]	0.35	0.07	0.40	0.69
	Public/Patients Preventing AMR	−0.69 [−1.47, 0.15]	0.37	−0.34	1.87	0.065
	Comp. Animal Vets Preventing AMR	−0.40 [−1.00, 0.15]	0.27	−0.23	1.51	0.14
	Pet Owners Preventing AMR	0.15 [−0.54, 0.87]	0.33	0.08	0.43	0.67
	Farm Vets Preventing AMR	0.10 [−0.81, 0.98]	0.41	0.05	0.26	0.80
	Farmers Preventing AMR	−0.09 [−0.87, 0.51]	0.34	−0.04	0.25	0.81

^1^*R^2^* = 0.052 for Block 1; Δ*R^2^* = 0.152 for Block 2 (*p* = 0.031). ^2^ BCa 95% CI = 95% bias-corrected and accelerated bootstrap confidence intervals. Bootstrap results are based on 1000 bootstrap samples. Comp. = Companion. V. = Versus.

**Table 6 antibiotics-10-00445-t006:** Vets’ Values.

Values	Mean	Lower BCa 95% CI ^1^	Upper BCa 95% CI ^1^	Standard Deviation
Hedonic	6.38	6.10	6.65	1.34
Egoistic	4.61	4.37	4.85	1.17
Altruistic	6.76	6.48	7.04	1.35
Biospheric	6.33	6.05	6.62	1.38

^1^ BCa 95% CI = 95% bias-corrected and accelerated bootstrap confidence intervals. Bootstrap results are based on 1000 bootstrap samples.

**Table 7 antibiotics-10-00445-t007:** Hierarchical Regression Model for Values as Predictors of Prescribing.

	Variable	*B* [BCa 95% CI] ^2^	*SE B*	*Β*	*t*	*p*
Block 1	Constant	3.20 [2.59, 3.79]	0.29		11.23	<0.001
	Control v. Economics	0.09 [−0.71, 1.01]	0.39	0.03	0.24	0.81
	Control v. Farmer	0.58 [−0.20, 1.43]	0.39	0.20	1.50	0.14
	Control v. Relationship	0.76 [−0.02, 1.56]	0.38	0.26	1.99	0.050
Block 2 ^1^	Constant	2.97 [0.33, 5.54]	1.20		2.48	0.015
	Control v. Economics	0.08 [−0.80, 1.12]	0.41	0.03	0.19	0.85
	Control v. Farmer	0.56 [−0.32, 1.44]	0.40	0.19	1.39	0.17
	Control v. Relationship	0.75 [−0.05, 1.60]	0.40	0.26	1.87	0.065
	Hedonic	−0.03 [−0.25, 0.19]	0.16	−0.03	0.23	0.82
	Egoistic	−0.01 [−0.33, 0.31]	0.14	−0.01	0.08	0.94
	Altruistic	0.02 [−0.26, 0.28]	0.12	0.02	0.18	0.86
	Biospheric	0.05 [−0.23, 0.31]	0.12	0.05	0.43	0.67

^1^*R^2^* = 0.249 for Block 1; Δ*R^2^* = 0.006 for Block 2 (*p* = 0.97). ^2^ BCa 95% CI = 95% bias-corrected and accelerated bootstrap confidence intervals. Bootstrap results are based on 1000 bootstrap samples. V. = Versus.

**Table 8 antibiotics-10-00445-t008:** Reliability Statistics for Values and Beliefs Measures.

Measure	Subscale	Cronbach’s *α*
Values	Hedonic	0.79
	Egoistic	0.66
	Altruistic	0.75
	Biospheric	0.88
Responsibility for causing AMR	Human medics	0.70
	Public/patients	0.56 ^1^
	Companion animal vets	0.83
	Pet owners	0.68 ^1^
	Farm animal vets	0.73
	Farmers	0.78
Responsibility for preventing AMR	Human medics	0.65 ^1^
	Public/patients	0.57 ^1^
	Companion animal vets	0.82
	Pet owners	0.74 ^1^
	Farm animal vets	0.73
	Farmers	0.81

^1^ Updated value of *α* after one item removed from subscale.

## Data Availability

Data available in a publicly accessible repository. The data presented in this study are openly available in UK Data Service at http://doi.org/10.5255/UKDA-SN-854821 kdj (accessed on 13 April 2021) (Data Collection title: Experimental vignette and cross-sectional survey with farm veterinarians, 2018). Demographic data have been removed from the publicly accessible file to ensure participant anonymity.

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
