# Peer review of "Examining the Effect of Context, Beliefs, and Values on UK Farm Veterinarians’ Antimicrobial Prescribing: A Randomized Experimental Vignette and Cross-Sectional Survey"

_antibiotics, 2021, doi:10.3390/antibiotics10040445_

Round 1
Reviewer 1 Report
The authors analyze the issue of antimicrobial treatment of animals in a very original way, taking into account not only clinical factors, but also factors such as psychological, social and environmental factors. This can make a significant contribution to helping veterinarians and livestock breeders address this pressing problem.
- The INTRODUCTION contains ample references to recent literature on the topic. The study brings a new perspective to the current field of knowledge about antimicrobial therapy in veterinary medicine.
- The RESULTS section adequately presents the authors' own results.
- Appropriate methods were used to statistically verification the results and were properly interpreted.
- The tables contain enough data to show the results of the study.
- DISCUSSION provides a correct interpretation of the study in the context of the literature and is a conclusion that is supported by the results obtained.
- Dear Authors, I advise you to put a Materials and Methods section before the Results section. This will make it easier for readers to analyze the results of the study.
Sincerely, Reviewer.
Author Response
REVIEWER 1 - Comments and Suggestions for Authors
|
Comment |
Response |
|
The authors analyze the issue of antimicrobial treatment of animals in a very original way, taking into account not only clinical factors, but also factors such as psychological, social and environmental factors. This can make a significant contribution to helping veterinarians and livestock breeders address this pressing problem. |
Thank you for this positive feedback on our study. |
|
The INTRODUCTION contains ample references to recent literature on the topic. The study brings a new perspective to the current field of knowledge about antimicrobial therapy in veterinary medicine. |
Thank you; we are pleased to hear our literature review is sufficiently comprehensive and that the study brings a novel perspective. |
|
The RESULTS section adequately presents the authors' own results. |
Thank you. |
|
Appropriate methods were used to statistically verification the results and were properly interpreted. |
Thank you. |
|
The tables contain enough data to show the results of the study. |
Good; we endeavoured to present data in sufficient detail |
|
DISCUSSION provides a correct interpretation of the study in the context of the literature and is a conclusion that is supported by the results obtained. |
Thank you. |
|
Dear Authors, I advise you to put a Materials and Methods section before the Results section. This will make it easier for readers to analyze the results of the study. |
Thank you for this suggestion; it is the journal’s house style to present the manuscript using this structure, so we have not amended. |
Reviewer 2 Report
GENERAL COMMENTS
The manuscript contains important observations on factors influencing farm veterinarian antimicrobial prescribing.
To improve readability and accessibility I recommend (as referred to below) that much of the current results section be included as a supplement, leaving an abbreviated summary and summary table in the body of the text.
The sample size sought and guided by power analysis was not achieved and this leads to an important limitation that is difficult to quantify. However, I believe the value of the observations could be increased by including Shannon information or S values (for example, see … Cole, S. R., J. K. Edwards and S. Greenland (2020). "Surprise!" American Journal of Epidemiology 190(2): 191-193 AND Rothman, K. J. (2020). "Rothman Responds to “Surprise!”." American Journal of Epidemiology 190(2): 194-195)
SPECIFIC COMMENTS
Lines 1-4 CHAGE OF TITLE
Examining the Effect of Context, Beliefs, and Values on Veterinarians’ Antimicrobial Prescribing: A Randomized Experimental Vignette and Cross-Sectional Survey OF UK FARM VETERINARIANS
Lines 11-26 ABSTRACT
As the abstract is often the sole source of information extracted from published studies I recommend that the abstract should mention the small sample size and possibility of false positive results.
Line 58 ALTERNATIVE EXPRESSION
‘claims for’ OR ‘knowledge of’
Line 80 ADD REERENCE
Doidge et al 2019 [10]
Line 121
‘…vets and doctors AND OTHERS’
Line 211 ABBREVIATIONS
Define mdr (is this median? Why not use mean?) and IQR (interquartile range?)
Line 222 FIGURE 1
For an international audience, why not replace bar diagram with a labelled map of the UK?
Line 247 ALTERNATIVE EXPRESSION
‘check’ OR ‘confirm that’
Line 267
There is no discussion of how to determine the effect size in the context of the analysis. Effect size is very important so some explanation is warranted.
I suggest that the Shannon information be provided, a more intuitive way to capture the analysis.
Line 274 FIGURE 4
What are the units of the Y axis and what do they mean – some context on effect size would be useful.
Line 295 TABLE 3
For each belief (Causing and Preventing) I would reorder the subscales in descending order of mean value
What are the units of measurement?
Lines 310 and 323 FIGURES 5 and 6
These figures are superfluous in consideration of the information in Table 3
Lines 254-411
For the non-biometrician the long explanation of the results set out in sections 2.2-2.6 could be summarised with one Table presented the results – for example a table with two columns, the first for hypothesis 1-10 and the second with the result and perhaps a third column with comments.
The long version of the results and the current non-superfluous tables and figures could be included in the supplementary data.
Line 489
It is not clear how the reference list was compiled but at least two additional references add information to supplement the discussion, for example …
Zhuo, A., M. Labbate, J. M. Norris, G. L. Gilbert, M. P. Ward, B. V. Bajorek, C. Degeling, S. J. Rowbotham, A. Dawson, K. A. Nguyen, G. A. Hill-Cawthorne, T. C. Sorrell, M. Govendir, A. M. Kesson, J. R. Iredell and D. Dominey-Howes (2018). "Opportunities and challenges to improving antibiotic prescribing practices through a One Health approach: results of a comparative survey of doctors, dentists and veterinarians in Australia." BMJ Open 8(3): e020439.
Norris, J. M., A. Zhuo, M. Govendir, S. J. Rowbotham, M. Labbate, C. Degeling, G. L. Gilbert, D. Dominey-Howes and M. P. Ward (2019). "Factors influencing the behaviour and perceptions of Australian veterinarians towards antibiotic use and antimicrobial resistance." PLoS One 14(10): e0223534.
Line 540
While no claims for representativeness of the sample of farm vets is made there is still is likely to be accessible information that would enable some comments to be made. For example, in view of the known demographics of vets in other countries, it seems highly unlikely that a sample of farm vets would be approximately equally divided between male and female.
Line 552
‘It is possible that this study was under-powered…’ OR ‘It is LIKELY that this study was under-powered…’
Line 651
There may be an economic argument not to prescribe antibiotics to lactating cows, however, this frequently leads to the use of ceftiofur which has no milk withholding period when administered to lactating dairy cattle. This suggests that other factors may have been associated with the amendment of the scenario.
Line 776
By how much did the drop out rate diminish?
Lines 817-830
The conclusion should mention the small sample size and possibility of false positives
Author Response
REVIEWER 2 - Comments and Suggestions for Authors
|
Comment |
Response |
|
GENERAL COMMENTS |
N/A |
|
The manuscript contains important observations on factors influencing farm veterinarian antimicrobial prescribing. |
Thank you for this positive summary. |
|
To improve readability and accessibility I recommend (as referred to below) that much of the current results section be included as a supplement, leaving an abbreviated summary and summary table in the body of the text. |
Thank you for this comment; we have decided to leave much of the results in the main manuscript but have moved some sentences from the results to the supplement. Please see our responses below for details. |
|
The sample size sought and guided by power analysis was not achieved and this leads to an important limitation that is difficult to quantify. However, I believe the value of the observations could be increased by including Shannon information or S values (for example, see … Cole, S. R., J. K. Edwards and S. Greenland (2020). "Surprise!" American Journal of Epidemiology 190(2): 191-193 AND Rothman, K. J. (2020). "Rothman Responds to “Surprise!” ." American Journal of Epidemiology 190(2): 194-195) |
Thank you for suggesting that we also include the S-values in our reporting. We have now included S-values throughout the results (all highlighted as changes) and have also added a section to the methods (lines 915-930). If you feel this section is unnecessary, or if our language in this section is imprecise, please do let us know and we will be happy to amend further. |
|
SPECIFIC COMMENTS |
N/A |
|
Lines 1-4 CHAGE OF TITLE Examining the Effect of Context, Beliefs, and Values on Veterinarians’ Antimicrobial Prescribing: A Randomized Experimental Vignette and Cross-Sectional Survey OF UK FARM VETERINARIANS |
Thank you for this suggestion – we have edited the title as follows: Examining the Effect of Context, Beliefs, and Values on UK Farm Veterinarians’ Antimicrobial Prescribing: A Randomized Experimental Vignette and Cross-Sectional Survey |
|
Lines 11-26 ABSTRACT As the abstract is often the sole source of information extracted from published studies I recommend that the abstract should mention the small sample size and possibility of false positive results. |
This is a good point. We have added this to the abstract as suggested (lines 25-27). |
|
Line 58 ALTERNATIVE EXPRESSION ‘claims for’ OR ‘knowledge of’ |
We have amended this sentence as suggested (line 67). |
|
Line 80 ADD REERENCE Doidge et al 2019 [10] |
Thank you for spotting this omission; we have added [10] to line 90. |
|
Line 121 ‘…vets and doctors AND OTHERS’ |
This sentence has been edited to add “and others” (line 137) |
|
Line 211 ABBREVIATIONS Define mdr (is this median? Why not use mean?) and IQR (interquartile range?) |
We reported median and interquartile range as these variables were not normally distributed. We have now defined the abbreviations in the text (lines 229-230). |
|
Line 222 FIGURE 1 For an international audience, why not replace bar diagram with a labelled map of the UK? |
We have explored the possibility of including a labelled map (based on open source maps), but have decided against this as we do not wish to accidentally include a potentially inaccurate figure. Instead, we have included a hyperlink to an official map of UK regions produced by the Office for National Statistics (see lines 242-244). |
|
Line 247 ALTERNATIVE EXPRESSION ‘check’ OR ‘confirm that’ |
Edited as suggested (line 267). |
|
Line 267 There is no discussion of how to determine the effect size in the context of the analysis. Effect size is very important so some explanation is warranted. I suggest that the Shannon information be provided, a more intuitive way to capture the analysis. |
We had originally commented that the p-value and effect size suggested there was no meaningful difference here. We have now edited this sentence to highlight that the observed effect size is considered small, and we have added the S-value as suggested (lines 287-289). |
|
Line 274 FIGURE 4 What are the units of the Y axis and what do they mean – some context on effect size would be useful |
The units are the responses on the 5-point Likert scale related to likelihood of prescribing antibiotics. We have added some explanation to the note for Figure 4 to make this clearer (lines 294-295). Please note we have also re-scaled the Y axis, so that the scale runs from 1-5 instead of 0-5 (as 1 was the lowest possible value; apologies for this oversight in the original figure). Effect size is commented on in the text (lines 287-289) |
|
Line 295 TABLE 3 For each belief (Causing and Preventing) I would reorder the subscales in descending order of mean value What are the units of measurement? |
Thank you for this suggestion; we have reordered Table 3 accordingly. The units are the responses on the 5-point Likert scale related to beliefs about how much each group’s actions contribute to causing/preventing AMR. We have added this explanation to the table note (lines 325-328). |
|
Lines 310 and 323 FIGURES 5 and 6 These figures are superfluous in consideration of the information in Table 3 |
We have removed Figures 5 and 6 as we agree they present the same information as in Table 3. We have, however, added two columns to Table 3 to include the p-values (that were referenced in the figures) and the newly calculated s-values. |
|
Lines 254-411 For the non-biometrician the long explanation of the results set out in sections 2.2-2.6 could be summarised with one Table presented the results – for example a table with two columns, the first for hypothesis 1-10 and the second with the result and perhaps a third column with comments.
The long version of the results and the current non-superfluous tables and figures could be included in the supplementary data. |
Thank you for this suggestion, however we would prefer to keep the long explanation of the results within the main manuscript so that the textual discussion of the findings and all the key metrics (e.g. means by conditions, test statistics, full regression models) are presented together.
We have, however, reviewed sections 2.2 to 2.6 to remove some of the technical discussion about the inferential tests and have moved this information to the supplementary file as suggested. |
|
Line 489 It is not clear how the reference list was compiled but at least two additional references add information to supplement the discussion, for example …
Zhuo, A., M. Labbate, J. M. Norris, G. L. Gilbert, M. P. Ward, B. V. Bajorek, C. Degeling, S. J. Rowbotham, A. Dawson, K. A. Nguyen, G. A. Hill-Cawthorne, T. C. Sorrell, M. Govendir, A. M. Kesson, J. R. Iredell and D. Dominey-Howes (2018). "Opportunities and challenges to improving antibiotic prescribing practices through a One Health approach: results of a comparative survey of doctors, dentists and veterinarians in Australia." BMJ Open 8(3): e020439.
Norris, J. M., A. Zhuo, M. Govendir, S. J. Rowbotham, M. Labbate, C. Degeling, G. L. Gilbert, D. Dominey-Howes and M. P. Ward (2019). "Factors influencing the behaviour and perceptions of Australian veterinarians towards antibiotic use and antimicrobial resistance." PLoS One 14(10): e0223534. |
The reference list was based on a more extensive literature review that was conducted as part of the first author’s doctoral research, with a smaller number of these papers being selected for inclusion in this manuscript; the references in this sentence were selected as together they covered examples of research conducted across a range of different groups.
However, we were not aware of these additional references, and they are indeed helpful in the discussion around groups’ beliefs about their own and others’ responsibility for AMR. We have therefore edited the discussion and included these two papers as references (lines 551-552 and 557-565). |
|
Line 540 While no claims for representativeness of the sample of farm vets is made there is still is likely to be accessible information that would enable some comments to be made. For example, in view of the known demographics of vets in other countries, it seems highly unlikely that a sample of farm vets would be approximately equally divided between male and female. |
Thank you for highlighting this. There are indeed some data available about the UK veterinary profession. We have therefore added to the limitations section a discussion of our study sample in relation to the UK vet population (lines 617-629) |
|
Line 552 ‘It is possible that this study was under-powered…’ OR ‘It is LIKELY that this study was under-powered…’ |
Edited as suggested (line 639) |
|
Line 651 There may be an economic argument not to prescribe antibiotics to lactating cows, however, this frequently leads to the use of ceftiofur which has no milk withholding period when administered to lactating dairy cattle. This suggests that other factors may have been associated with the amendment of the scenario. |
In amending the scenario (to change the description from a ‘dairy cow’ to a ‘dry cow’) we were aiming to remove potential ambiguity from the vignette; we wanted participants to be presented with clinical information that was as ‘neutral’ as possible, and not be presented with a situation in which economics might be explicitly or implicitly influencing the prescribing decision - if participants assumed the cow was lactating, then the economic argument for no antibiotics, (or as you correctly point out, for a product with nil milk withhold) might come into play, and we explicitly wanted to test the effect of economics in one of the vignette conditions, as well as have a ‘clinical information only’ control vignette. If there was potentially an economic factor embedded in the control version of the vignette, this would contaminate both the control vignette and the other experimental versions of the vignette.
We also realised that if we did not explicitly describe the cow as ‘dry’ then it would have left open the possibility that some participants would imagine the cow was lactating while others might imagine she was dry, further confounding the results.
We have therefore added to this section to include these points and make our reasoning for amending the description of the cow clearer (lines 752-763). |
|
Line 776 By how much did the drop out rate diminish? |
It reduced from 40% to 24%. This information has been added to line 887. |
|
Lines 817-830 The conclusion should mention the small sample size and possibility of false positives |
We have added a sentence in this regard (lines 954-956) |
Reviewer 3 Report
I thought the paper was very clearly presented and was comprehensive in its inclusion of literature. It's a shame you didn't get the sample size you wanted, but I think you discuss your statistical methods and results in enough detail to show the value of study to readers.
Some minor comments:
I didn’t understand this sentence in the abstract: "However, when vets’ beliefs about groups’ responsibilities for preventing AMR were accounted for, exposure to the relationship condition of the vignette resulted in increased likelihood of prescribing antibiotics." I don’t think the reader will understand the meaning of ‘relationship condition of the vignette’. Could you express it more clearly?
Line 50 I’m not sure if this statement is justified: “In contrast, although the emphasis is beginning to shift, much of 50 the published research exploring prescribing in veterinary medicine, has focused either 51 on patterns of use [27–30] or on clinical factors influencing prescribing, such as underlying 52 animal health or the use of antimicrobial susceptibility testing [31–33].” You have listed and go on to list lots of research on context, beliefs and values. I’d suggest presenting the novelty of your study differently. Is the novelty in the vignette methodology? And that the vignettes build on previous research.
In the limitations could you include something about how there might be differences between what vets would say they’ll do in the vignette situations and what they might actually do in reality. Because of lack of self-knowledge, differences in how different people define or understand the vignette scenarios in reality etc.
Author Response
REVIEWER 3 - Comments and Suggestions for Authors
|
Comment |
Response |
|
I thought the paper was very clearly presented and was comprehensive in its inclusion of literature. It's a shame you didn't get the sample size you wanted, but I think you discuss your statistical methods and results in enough detail to show the value of study to readers. |
Thank you for these positive comments on our paper. Yes, we were also disappointed that we were not able to recruit as many participants as planned for, but we are pleased to hear that our write-up of the methods and results has been sufficient to nonetheless demonstrate the value of the study. |
|
Some minor comments: |
N/A |
|
I didn’t understand this sentence in the abstract: "However, when vets’ beliefs about groups’ responsibilities for preventing AMR were accounted for, exposure to the relationship condition of the vignette resulted in increased likelihood of prescribing antibiotics." I don’t think the reader will understand the meaning of ‘relationship condition of the vignette’. Could you express it more clearly? |
We have reworded this sentence, but please advise if you think this is still not clear (lines 22-24). |
|
Line 50 I’m not sure if this statement is justified: “In contrast, although the emphasis is beginning to shift, much of 50 the published research exploring prescribing in veterinary medicine, has focused either 51 on patterns of use [27–30] or on clinical factors influencing prescribing, such as underlying 52 animal health or the use of antimicrobial susceptibility testing [31–33].” You have listed and go on to list lots of research on context, beliefs and values. I’d suggest presenting the novelty of your study differently. Is the novelty in the vignette methodology? And that the vignettes build on previous research. |
Thank you for raising this point. We were trying to emphasise that compared to human medicine (where there is a large body of both primary empirical studies and review articles) there is relatively little research in veterinary medicine that has explored the non-clinical aspects of treatment decisions.
We have therefore edited the paragraph you have referred to, to highlight this relative difference between the human and veterinary evidence bases (lines 55-59, and line 62). We have also edited lines 80-82 to more explicitly relate the vignette methodology to the existing evidence base in veterinary medicine, and to highlight that vignettes are relatively under-utilized in veterinary medicine |
|
In the limitations could you include something about how there might be differences between what vets would say they’ll do in the vignette situations and what they might actually do in reality. Because of lack of self-knowledge, differences in how different people define or understand the vignette scenarios in reality etc. |
You are correct to point out that the vignette responses are only proxies for actual behaviour, and that people may not behave in everyday clinical practice in the way that they indicate in response to the vignettes. We have therefore added an additional paragraph to the limitations section to address this point (lines 674-689). |